# Human γδ T Cell Subsets and Their Clinical Applications for Cancer Immunotherapy

**DOI:** 10.3390/cancers14123005

**Published:** 2022-06-18

**Authors:** Derek Lee, Carl J. Rosenthal, Natalie E. Penn, Zachary Spencer Dunn, Yang Zhou, Lili Yang

**Affiliations:** 1Department of Microbiology, Immunology & Molecular Genetics, University of California, Los Angeles, CA 90095, USA; ylee932@g.ucla.edu (D.L.); nepenn168@g.ucla.edu (N.E.P.); zacharsd@usc.edu (Z.S.D.); zzydcat@ucla.edu (Y.Z.); 2Molecular Biology Institute, University of California, Los Angeles, CA 90095, USA; 3Department of Molecular, Cell and Developmental Biology, University of California, Los Angeles, CA 90095, USA; cjr@dt3.org; 4Mork Family Department of Chemical Engineering and Materials Science, University of Southern California, Los Angeles, CA 90089, USA; 5Eli and Edythe Broad Center of Regenerative Medicine and Stem Cell Research, University of California, Los Angeles, CA 90095, USA; 6Jonsson Comprehensive Cancer Center, David Geffen School of Medicine, University of California, Los Angeles, CA 90095, USA

**Keywords:** gamma delta T (γδ T) cells, cancer immunotherapy, chimeric antigen receptor T (CAR-T) cells, allogeneic cell therapy, butyrophilins (BTN), zoledronate (ZOL)

## Abstract

**Simple Summary:**

Research into the immunotherapeutic potential of T cells has predominantly focused on conventional alpha beta (αβ) T cells, which recognize peptide antigens presented by polymorphic major histocompatibility complex (MHC) class I and class II molecules. However, innate-like T cells, such as gamma delta (γδ) T cells, also play important roles in antitumor immunity. Here, we review the current understanding of γδ T cells in antitumor immunity and discuss strategies that could potentially maximize their potential in cancer immunotherapy.

**Abstract:**

Gamma delta (γδ) T cells are a minor population of T cells that share adaptive and innate immune properties. In contrast to MHC-restricted alpha beta (αβ) T cells, γδ T cells are activated in an MHC-independent manner, making them ideal candidates for developing allogeneic, off-the-shelf cell-based immunotherapies. As the field of cancer immunotherapy progresses rapidly, different subsets of γδ T cells have been explored. In addition, γδ T cells can be engineered using different gene editing technologies that augment their tumor recognition abilities and antitumor functions. In this review, we outline the unique features of different subsets of human γδ T cells and their antitumor properties. We also summarize the past and the ongoing pre-clinical studies and clinical trials utilizing γδ T cell-based cancer immunotherapy.

## 1. Introduction

Chimeric antigen receptor (CAR) engineered T cell (CAR-T) therapy has revolutionized the treatment of hematological malignancies over the last decade with approved therapies for leukemias, lymphomas, and multiple myeloma (MM). Despite significant improvements in clinical outcomes, CAR-T therapy has faced some hurdles, particularly in treating solid tumors [1]. In this regard, cells with distinct immunological features, such as natural killer (NK) cells, NKT cells, and γδ T cells, offer alternative cell sources for CAR engineering to treat cancer. Based on the surface expression of αβ or γδ T cell receptors (TCRs), T cells are classified into two primary subpopulations—αβ T and γδ T cells. While the majority of T cell research and therapeutic application has focused on αβ T cells, γδ T cells are also significant participants in cancer immunity. In fact, Gentles et al. studied complex associations between 22 different leukocyte subsets and cancer survival across 5782 clinical tumor samples and found that intra-tumoral γδ T cells were the best predictors of overall patient survival [2]. αβ T cells are adaptive immune cells that recognize their target antigens presented on the major histocompatibility complex (MHC) molecules. γδ T cells, on the other hand, are innate immune cells that function in an MHC-unrestricted manner and are able to recognize surface proteins such as butyrophilin (BTN) and CD1 molecules. In addition, γδ T cells have a low risk of developing graft-versus-host disease (GvHD) in allogeneic adoptive transfer settings [3,4], and thus provide a new opportunity for “off-the-shelf” cellular therapeutics. Past studies have shown that γδ T cells exert potent cytotoxicity against various types of tumors [5,6,7,8]. However, compared with their αβ T cell counterparts, they only comprise 1–5% of total T cells [9]. In this review, we will first provide an introduction to γδ T cell biology and then summarize past and current clinical trials utilizing γδ T based-immunotherapy. We will also discuss the challenges of γδ T based-immunotherapy and the future directions that may overcome these obstacles.

## 2. γδ T Cell Subsets and Their Key Features

Recognizing the physiological differences between murine and human γδ T cells has been critical for understanding their immunotherapeutic potential. Rather than an orthologous conservation between species, mice and humans each possess distinct γδ T cell subtypes with analogous but not precisely identical functions [10]. Mouse subtypes are typically defined by the γ chain of the γδ TCR, while human γδ T cell subtypes are typically defined by the δ chain. The most relevant subtypes for mice are accordingly Vγ1-7, and those for humans are Vδ1-3 [10].

The human γδ T subsets that have been most comprehensively studied are Vδ1+ and Vδ2+ T cells, both of which arise during fetal development. Shortly after birth, Vδ1+ T cells are the primary subtype, and occupy the cord blood, postnatal thymus, gut, and skin [11,12,13,14,15]. Adults also maintain a population of Vδ1+ T cells. Although they make up only about 10–15% of γδ T cells in the blood, they reside primarily in peripheral tissues such as gut epithelia, dermis, spleen, and liver [16,17]. While the development of Vδ1+ T cells is not fully elucidated, Vδ2+ T cell development has been characterized in greater detail. TCR Vδ2 chains are frequently paired with Vγ9 chains during development [18]. The cells first arise in the fetus, at around five to seven weeks, then migrate from the fetal liver to the thymus at 8 weeks [19]. Though Vδ1+ T cells outnumber Vδ2+ cells immediately after birth, the Vδ2+ cells expand to predominance in the blood and spleen by adulthood, comprising 80–85% of total peripheral γδ T cells [16]. The Vγ9 chain is germline-encoded to recognize target BTN proteins, suggesting that BTN family molecule exposure may not be necessary during maturation [20,21]. However, variation in the Jδ chain of fetal and adult Vδ2 cells indicates that γδ T cell expansion during maturation to adulthood also requires antigen stimulation and selection [20,21]. Vδ3+ T cells comprise the majority of the remaining γδ T population [22]. Their lineage and rearrangement are less well described than the other subtypes, though they are known to complex with Vγ2 and Vγ3 for recognizing CD1d-presented antigens [22]. These cells are found in peripheral blood and the liver, and have been detected in patients with B-cell leukemia as well as in patients with human immunodeficiency virus (HIV) and cytomegalovirus (CMV) infections [22]. Other human γδ T cell subsets such as Vδ4, Vδ6, Vδ7, and Vδ8 T cells have been found in the peripheral blood of lymphoma patients, but their functions have yet to be determined [23].

## 3. γδ T Cell Recognition and Killing of Tumor Cells

γδ T cells that express the γ9 chains paired with the δ2 chains (Vγ9Vδ2 T cells) are the major γδ T cell population in human peripheral blood. The Vγ9Vδ2 T cells react to cells with accumulated intracellular phosphoantigens (pAgs), intermediate metabolites produced by infected or transformed cells. These metabolites include isopentenyl pyrophosphate (IPP) formed by the mevalonate (MVA) pathway of tumor cells, and (E)-4-Hydroxy-3-methyl-but-2-enyl pyrophosphate (HMB-PP) produced by microbial isoprenoid biosynthesis [24]. Overproduction of IPP in cancer cells as a result of dysregulated MVA pathway leads to activation of Vγ9Vδ2 T cells. A past study showed that oncogenic mutations cause the tumor suppressor protein p53 to interact with sterol regulatory element–binding protein-2 (SREBP-2) nuclear transcription factor, which augments the expression of MVA pathway genes [25,26]. While the precise TCR targeting mechanism of Vγ9Vδ2 T cells has been controversial, recent evidence has shown that the TCR Vγ9 chain directly recognizes BTN2A1, which is associated with BTN3A1 (CD277), another ligand recognized by the TCR Vδ2 chain [27] (Figure 1). BTNs belong to the immunoglobulin superfamily and are structurally similar to B7 family molecules, which include CD80, CD86, and programmed cell death-1 ligand (PD-L1) [28]. pAgs bind the intracellular B30.2 domain of BTN3A1, which enables the BTN2A1-BTN3A1 complex to bind the Vγ9Vδ2 TCR [27]. Strategies to expand Vγ9Vδ2 T cells thus focus on inducing pAg accumulation for TCR activation. Synthetic phosphoantigen analogues mimicking the pAg accumulated state, such as bromohydrin pyrophosphate (BrHPP) and 2-methyl-3-butenyl-1-pyrophosphate (2M3B1PP), can directly stimulate Vγ9Vδ2 T cells [29,30]. Stimulation can also be achieved with bisphosphonates, a class of drugs that prevent or slow down bone loss, such as zoledronate (ZOL) and pamidronate (PAM) [30,31,32,33,34]. These small molecule compounds inhibit farnesyl pyrophosphate synthase in the MVA metabolic pathway, leading to pAg accumulation in treated cells which activates the Vγ9Vδ2 TCR [24]. Notably, the Vγ9Vδ2 T cell subpopulation still retains diversity between and within individuals, and therefore cloning of a single, high-affinity Vγ9Vδ2 TCR to create an effector population could produce more effective therapies than the use of a polyclonal existing population [35].

Non-Vγ9Vδ2 T cells have more diverse TCRs without pAg-mediated BTN recognition capacity. Instead, these non-Vγ9Vδ2 TCRs recognize other molecules such as the CD1 family of surface glycoproteins presenting lipid antigens [36] (Figure 1). Vδ1+ and Vδ3+ subsets have been found to recognize CD1 proteins, although recognition of lipid antigens on CD1 molecules is typically associated with NKT cells [36,37,38,39,40]. However, unlike BTN molecules, which are commonly expressed on both hematologic and solid tumors [41], CD1+ tumor cells mostly arise from myelomonocytic and B-cell lineages, and only a few solid tumors have been discovered to be CD1+ [42]. While it is known that CD1d recognition by NKT cells enables their antitumor response, this reactivity in non-Vγ9Vδ2 T cells remains to be demonstrated [43].

Besides their TCRs, human γδ T cells also express other surface molecules that can recognize tumor cells and provide additional stimulatory signals. Natural killer receptors expressed by γδ T cells–including Natural Killer Group 2D (NKG2D), DNAX Accessory Molecule-1 (DNAM-1), NKp30, NKp44, and NKp46–enable them to be activated by and target malignantly transformed cells, which often overexpress ligands for these receptors [44,45,46,47,48,49] (Figure 1). Both hematologic and solid tumors typically express MHC class I chain-related polypeptide A (MICA) and MICB, as well as members of the UL16 binding protein (ULBP) family (ULBP1-6), which activate NKG2D-expressing Vδ1+ and Vδ2+ T cells. The binding of DNAM-1 to its target CD155 (poliovirus receptor, PVR) [50] or CD112 (Nectin-2) [51] can promote the NK cell-mediated elimination of transformed and virus-infected cells. Although both NKG2D and DNAM-1 are expressed on Vδ1+ and Vδ2+ T cells, NKp30, NKp44, and NKp46 are only expressed on Vδ1+ T cells [52].

The antitumor cytotoxicity of γδ T cells occurs through activation induced perforin and granzyme B release, which causes pore formation, granzyme protease entry, and subsequent apoptosis of the tumor cells [53] (Figure 2). In addition to the perforin-granzyme pathway, γδ T cells also kill tumor cells by expressing TNF-related apoptosis-inducing ligand (TRAIL) and Fas ligand (FasL), which engage their respective receptors expressed by tumor cells to induce apoptosis [53,54,55]. Furthermore, γδ T cells can target tumors through antibody dependent cellular cytotoxicity (ADCC), whereby CD16 (Fcγ receptor III) expressed on γδ T cells binds the Fc region of antibodies bound to a target cell, leading to tumor lysis [53,56,57] (Figure 1). The Vγ9Vδ2 subset in particular has been shown to upregulate CD16 after stimulation, and effectively performs ADCC against tumors coordinated by monoclonal antibodies against HER2 (Trastuzumab) and CD20 (Rituximab) [57].

## 4. γδ T Cell Activation via Toll-like Receptors

Another co-stimulator is the class of toll-like receptors (TLRs), which can activate γδ T cells directly or by participating in DC activation [58]. TLRs are integral membrane glycoproteins which function as prototype pattern recognition receptors, dimerizing in response to specific lipid, peptide, or nucleic acid PAMPs (pathogen-associated molecular patterns) and DAMPs (danger-associated molecular patterns from damaged tissues) [58,59]. Binding induces signal transduction pathways upregulating the expression of inflammatory cytokines such as TNF-α, IL-6, IL-1β, IL-12, and IFN-γ [58,59]. However, TLR signaling can activate or suppress the immune system: with nine TLRs expressed in humans, the result of signaling can depend on the specific TLR and surrounding circumstances. For instance, several studies [60,61] indicated that TLR2 stimulation suppressed Treg potential by suppressing FOXP3, whereas TLR5 stimulation increased FOXP3 expression.

In immunotherapy, the use of TLR agonists enhances γδ T cell cytotoxicity and proliferation in a DC-dependent and independent manner [58] (Figure 1). γδ T cells have been shown to respond to a combination of TLR and TCR stimulation while influencing the antigen-presenting property of DCs in return [62,63]. DCs incubated and activated with γδ T cells increased expression of CD86 and MHC I, enhanced the production of IFN-γ by alloreactive T cells, and produce IL12p70 [62], a messenger which activates NK cells and promotes γδ T cell proliferation. When activating DCs through TLR signaling, γδ T cells are co-activated, leading to an increase in CD25 and CD69 expression as well as the production of inflammatory cytokines [64]. Vδ2+ cells, more so than Vδ1+ cells, have a strong IFN-γ production response to a variety of TLRs [65]. Vδ2+ T cells expanded using IPP plus resiquimod, an agonist for TLRs 7 and 8, and were shown to have increased antitumor cytotoxicity and lower PD-1 expression than Vδ2+ T cells expanded with IPP or ZOL [66]. Resiquimod led to the expansion of Vδ2+ T-cells with and without the aid of antigen-presenting cells (APCs), and decreased the presence of inhibitory APCs and downregulated their expression of immunosuppressive PD-L1 and CTLA-4 [66].

## 5. γδ T Cell Modulation of Tumor Microenvironment

The tumor microenvironment (TME) is composed of cancer cells, surrounding stroma including fibroblasts and vasculature, and infiltrating immune cells [67]. Although hypoxia and immunosuppressive conditions in the TME hinder many forms of cell therapy [67], γδ T cells may be more effective in this challenging environment due to their recognition of stress markers expressed on many tumors, MHC independent activation, production of inflammatory cytokines, and the ability to stimulate other immune cells for a coordinated antitumor response. In addition, it was reported that γδ T cells can target M1 and M2 macrophages [68]. Macrophages that infiltrate the TME may encounter microenvironment specific chemokines, growth factors, and other signals that polarize them away from an antitumor phenotype and towards a tumor promoting M2 state (tumor-associated macrophages, TAMs) [69]. TAMs are a significant component of the TME that support tumor growth and induce extracellular matrix (ECM) remodeling facilitating metastasis through factors such as VEGF and MMP-9 [70]. TAMs also inhibit immune cell antitumor activity by secreting immunosuppressive cytokines such as IL-10 and TGF-β and engaging inhibitory receptors on infiltrating lymphocytes [71]. In treating solid tumors, it is therefore beneficial to repolarize TAMs to an antitumor state, or to directly target and deplete TAMs.

While typically used to stimulate Vγ9Vδ2 T cells, ZOL is also useful against TAMs: it reduces TAM’s pro-tumoral activities, repolarizes TAMs to an antitumor state, and may guide Vγ9Vδ2 T cell killing of TAMs. Macrophages have been shown to respond to ZOL in vitro and in vivo [72,73], with increased unprenylated Rap1A levels indicating MVA pathway inhibition [72]. Tumors from ZOL treated mice had reduced vascularization and TAM numbers, as well as decreased VEGF and IL-10 and increased proinflammatory IFN-γ [73]. Furthermore, macrophages from these ZOL treated mice repolarized to an antitumor phenotype [73]. In other investigations, ZOL similarly reduced TAM MMP-9 expression while increasing TAM expression of cytokines that restore an antitumor macrophage state [74]. While ZOL alone has promising effects on TAMs, it offers further benefits when administered in combination with Vγ9Vδ2 T cells, where it facilitates direct macrophage targeting; phosphoantigen accumulation in treated macrophages enables Vγ9Vδ2 T cells to respond to these cells and kill them in a perforin-dependent manner [68].

## 6. γδ T Cell Interaction with Other Immune Cells

In addition to altering the TME, γδ T cells have further immunomodulatory roles in interactions with other immune cell types (Figure 2). Vγ9Vδ2 T cells act as antigen presenting cells (APCs) in a similar capacity to DCs [75]. Mature DCs (mDCs) are the canonical professional APC, and their ability to present tumor associated antigens to T cells makes them a crucial component of the anti-cancer immune response [76]. Vγ9Vδ2 T cells similarly act as APCs upon activation, upregulating MHC I and MHC II molecules and expressing APC associated adhesion and costimulatory molecules [75,77]. These cells also express the scavenger receptor CD36, which may facilitate their ability to uptake antigens from both apoptotic and live tumor cells [78]. After taking in tumor antigens, Vγ9Vδ2 T cells present those antigens to CD4+ αβ T cells and also utilize insulin-regulated aminopeptidase (IRAP) [79] to cross present them on MHC I molecules to prime CD8+ αβ T cells–a hallmark of professional APCs [75,77]. Antigen presentation by Vγ9Vδ2 T cells induces naive CD4+ and CD8+ αβ T cell differentiation and proliferation to a similar degree as DCs [75,77].

Beyond acting as APCs to activate T cells, γδ T cells have diverse roles interacting with other immune cells. One well characterized set of interactions is the bidirectional crosstalk between γδ T cells and DCs [80]. Vγ9Vδ2 T cells assist with DC maturation by producing TNF-α and IFN-γ, yielding DCs with upregulated co-stimulatory molecules and enhanced cytokine secretion which effectively induce αβ T cell proliferation and IFN-γ production [81]. Conversely, DCs can activate γδ T cells. CD86-CD28 cell contact interactions [64], as well as production of stimulatory cytokines such as IL-12, IL-1β, TNF-α, and type 1 IFNs [82,83,84,85], enable DCs to activate γδ T cell cytokine production and cytotoxicity. In particular, DCs treated with ZOL are especially effective at activating Vγ9Vδ2 T cells. ZOL pulsed DCs induced greater γδ T cell proliferation and expanded CD62L+ populations compared to untreated DCs [86,87]. Furthermore, Vγ9Vδ2 T cells activated by ZOL treated immature DCs had upregulated CD40L expression and IFN-γ production, and subsequently enhanced antitumor CD8+ αβ T cell proliferation [88].

γδ T cells also interact with and activate NK cells. Maniar et al. showed that IPP expanded Vγ9Vδ2 T cells co-cultured with NK cells increased NK cell cytotoxicity against tumors that are usually resistant to NK cytolysis [89]. These γδ T cells activate NK cells through co-stimulatory interactions, with CD137L presented on activated γδ T cells and CD137 expressed on activated NK cells. Activated NK cells augment their NKG2D expression and increase their tumor cell cytotoxicity. In addition, when stimulated with ZOL, Vγ9Vδ2 T cells also enhance NK tumor cell killing through ADCC pathways [89]. Further evidence that γδ T cells support NK cells comes from in vivo results suggesting a critical role for γδ T cells in enabling NK cell IFN-γ production during early antibacterial responses [90]. Additional work found that γδ T cells enhanced NK cellmediated IFN-γ production in early stages after ZOL stimulation, and these effects were formed by interactions between γδ T cells, DC-like cells, and NK cells [91].

## 7. Autologous γδ T Cell Adoptive Transfer Therapy

Early clinical investigations of γδ T cells predominantly used them autologously (Table 1 and Table 2). The first pilot clinical study of γδ T cells was conducted by Wilhelm et al. in 2003 [92]. Nineteen patients with relapsed or refractory (R/R) MM and low-grade non-Hodgkin lymphoma (NHL) were given intravenous (i.v.) IL-2 and PAM. Several patients experienced low-grade adverse responses to IL-2 infusion, and two experienced a grade 3 response, but none were severe enough to stop treatment. γδ T cells did not measurably proliferate in an initial cohort of ten patients, and the group did not experience any objective tumor response. An additional nine patients were selected based on in vitro γδ T cell proliferation in response to PAM/IL-2. In this cohort, five patients were found to have significant in vivo γδ T cell proliferation after treatment. Of the nine patients, three had partial remission and two additional patients had stable disease. This study reveals the interpatient heterogeneity of γδ T cell responsiveness to bisphosphonate compound in vivo and highlights the value of prescreening γδ T cells in vitro before autologous therapeutic use. A later phase I clinical trial conducted by Dieli et al. was the first to evaluate combined ZOL and IL-2 and confirmed the importance of IL-2 for maintaining γδ T cells in vivo [55]. They compared the effect of giving ZOL alone or in combination with IL-2 on γδ T cells in eighteen patients with metastatic hormone-refractory prostate cancer. All patients tolerated treatment well: eight patients across both groups developed expected, mild, and transient flu-like symptoms controlled by paracetamol, and two developed mild erythema around the IL-2 injection site, but no further adverse effects were observed. In nine patients receiving ZOL/IL-2, γδ T cells proliferated and had an effector memory phenotype, with IFN-γ, TRAIL, and perforin production. Three of these patients had partial remission, and five experienced stable disease. In contrast, γδ T cells from nine patients receiving only ZOL generally did not proliferate and had lower TRAIL serum concentrations. In this group, only one patient experienced partial remission, and one patient had stable disease.

The benefit of combined ZOL/IL-2 in vivo was further supported by another phase 1 clinical trial in 2010 by Meraviglia et al., who evaluated this treatment in ten terminal patients with advanced metastatic breast cancer [93]. The combination was well-tolerated: five patients developed expected, mild, and transient flu-like symptoms controlled by paracetamol, and two developed mild erythema around the IL-2 injection site, but no further adverse effects were observed. γδ T cells in three patients proliferated into an effector population with increased IFN-γ production. These patients had reduced CA15-3 tumor marker levels, concomitant with one instance of partial remission and two cases of stable disease. Patients without sustained γδ T cell numbers had progressively worsening disease. This study, taken in combination with previous results, supports a correlation between γδ T cell responsiveness and cancer prognosis.

Beyond supplementing endogenous γδ T cells with stimulatory molecules and cytokines, clinical studies have also assessed the autologous use of γδ T cells expanded ex vivo (Table 2). The first such investigation, a 2007 pilot study by Kobayashi et al., expanded γδ T cells from the peripheral blood of seven advanced renal cell carcinoma (RCC) patients [94]. γδ T cells were cultured with pAg stimulation, and then reinfused into patients along with IL-2. Antitumor effects were reported in five patients, three of which showed demonstrably slower tumor growth, but the other two showed increased growth speed. No patients experienced high-grade adverse effects, though many of them had low-grade (mild or bothersome) adverse effects from IL-2 infusions. Following up their pilot study, Kobayashi et al. took the ex vivo expansion, IL-2-supplemented system and added injection with ZOL several hours before cell infusion for a phase I/II study [95]. Of eleven patients with advanced RCC, one experienced a complete response and five experienced stable disease, but the rest continued to show disease progression. All patients experienced adverse effects spanning grades 1 through 4, but only two patients dropped out of the study, and all others saw a reduction in adverse events as their treatment continued. Compared to previous results, ex vivo stimulation in combination with ZOL and IL-2 significantly improved γδ T cytotoxicity and therapeutic effects.

One autologous study by Wada et al. identified cell localization as a potential issue to address, and therefore attempted localized injection of the therapeutic product [96]. Seven patients with gastric adenocarcinoma and secondary malignant ascites received ex vivo expanded γδ T cells injected directly into their peritoneal cavities. Adverse effects were limited to fever and ZOL-induced hypocalcemia, both of which were reversible and mild. One patient in the study experienced reduced ascites volume, and for another patient ascites almost completely disappeared from the peritoneal cavity.

**Table 1 cancers-14-03005-t001:** Autologous, in vivo expanded γδ T cell clinical trials.

Year	Author	Phase	Tumor Type	Treatment	Clinical Outcome
2003	Wilhelm et al. [92]	Pilot clinical study	MM, CLL, MZL	PAM + IL-2	**Cohort A *SD: 1/10, PD: 8/10*** - 1 patient dropped out of the study. - Little to no γδT proliferation was observed. **Cohort B *PR: 3/9, SD: 2/9, PD: 4/9*** - γδT proliferation was observed in 5/9 patients.
2007	Dieli et al. [55]	I	Prostate cancer	ZOL or ZOL + IL-2	**ZOL only *PD: 3/9, SD: 1/9, PR: 1/9*** - 4 patients dropped out of the study. - 1/4 that could be evaluated had a sustained effector population, and was the only patient with a PR. **ZOL + IL-2. *PD: 1/9, SD: 4/9, PR: 2/9*** - 2 patients dropped out of the study. - 5/7 had an increased effector population.
2010	Meraviglia et al. [93]	I	Breast cancer	ZOL + IL-2	** *PD: 4/10, SD: 2/10, PR: 1/10* ** - 3 patients dropped out of the study. - Each patient with SD or PD showed sharp decline in their γδ T cell population either before death or before reduction to SD.
2010	Bennouna et al. [97]	I	Solid tumor variety	BrHPP + IL-2	** *PD: 16/28, SD: 12/28* **
2012	Kunzmann et al. [98]	I/II	RCC, MM, AML	ZOL + IL-2	** *PR: 2/21, SD: 6/21, PD: 12/21* ** - 1 patient dropped out of the study.
2011	Lang et al. [99]	Pilot clinical study	RCC	ZOL + IL-2	** *SD: 5/12* ** - 3 patients dropped out of the study.
2016	Pressey et al. [100] (NCT01404702)	I	Neuroblastoma	ZOL + IL-2	** *PD: 4/4* **
2022	LAVA Therapeutics (NCT04887259)	I/IIa	CLL, MM	Bispecific Ab for Vγ9Vδ2 TCR and CD1d	** *No patient response has yet been reported.* ** - The therapy was safe and well-tolerated.

**Abbreviations** are as follows: **PD:** progressive disease; **SD:** stable disease; **PR:** partial response; **Ab:** antibody; **ZOL:** zoledronate/zoledronic acid; **PAM:** pamidronate; **BrHPP:** bromohydrin pyrophosphate; **AML:** acute myeloid leukemia; **CLL:** chronic lymphocytic leukemia; **MM:** multiple myeloma; **MZL**: marginal zone lymphoma; **RCC:** renal cell carcinoma.

**Table 2 cancers-14-03005-t002:** Autologous, ex vivo expanded γδ T cell clinical trials.

Year	Author	Phase	Tumor Type	Effector Cells	Clinical Outcome
2007	Kobayashi et al. [94] (NCT00588913)	Pilot clinical study	RCC	Vγ9Vδ2 T + IL-2	***PD: 7/7*** 5/7 showed an increased γδ T cell population.
2008	Bennouna et al. [101]	I	RCC	Vγ9Vδ2 T + IL-2	At one time point, ***SD: 6/10, PD: 4/10***. However, 4 patients were later withdrawn prematurely.
2009	Abe et al. [102]	Pilot clinical study	MM	Vγ9Vδ T	** *PD: 6/6* ** - The effector memory Vγ9+ T population increased, as well as the overall Vγ9+ T percentage in bone marrow and number in peripheral blood. - Expression of NKG2D and CD69 increased on Vγ9+ T cells after stimulation. - Increased IFN-γ production.
2010	Nakajima et al. [103] (C000000336)	I	NSCLC	Vγ9Vδ2 T	At one time point, ***SD: 3/10, PD: 5/10***. However, only 6 patients remained at the end of the study.
2011	Kobayashi et al. [95] (NCT00588913)	I/II	RCC	Vγ9Vδ2 T + ZOL + IL-2	** *PD: 5/11, SD: 5/11, CR: 1/11* **
2011	Sakamoto et al. [104] (C000000336)	I	NSCLC	Vγ9Vδ2 T	** *PD: 6/15, SD: 6/15* ** - 3 patients dropped out of the study.
2011	Nicol et al. [105]	I	Solid tumors	Vγ9Vδ2 T + ZOL /Vγ9Vδ2 T + ZOL + conventional therapy	Vγ9Vδ2 T+ ZOL only: ***SD: 3/15, PD: 11/15*** Vγ9Vδ2 T+ ZOL + other treatments: ***CR: 1/3, PR: 2/3*** - 1 patient dropped out of the study.
2013	Izumi et al. [106] (UMIN000000854)	-	CRC	Vγ9Vδ2 T	** *PD: 6/6* ** - All patients demonstrated expansion of the Vγ9Vδ2 T population.
2014	Kakimi et al. [107] (C000000336)	I	NSCLC	Vγ9Vδ2 T	** *PD: 6/15, SD: 6/15* ** - 3 patients dropped out of the study. - Increased IFN-γ release was observed. - Interestingly, Vγ9Vδ2 T population expansion did not correlate with prognosis.
2014	Wada et al. [96] (UMIN000004130)	Pilot clinical study	Gastric cancer	Vγ9Vδ2 T + ZOL	** *PD: 7/7* ** - Reduction in tumor ascites.
2015	Cui et al. [108]	Pilot clinical study	Gastric cancer	Chemotherapy + γδ T/NK/Killer cells	- Progression-free survival of the group infused with effectors was significantly higher than those treated with chemotherapy alone: 70% vs. 46.4% survived over the two years, respectively.
2017	Gadeta (NL6357)	I	AML, MM, MDS	Vγ9Vδ2 TCR transduced αβ T cells	-

**Abbreviations** are as follows: **PD:** progressive disease; **SD:** stable disease; **PR:** partial response; **CR:** complete response/remission; **ZOL:** zoledronate/zoledronic acid; **AML:** acute myeloid leukemia; **CRC:** colorectal cancer; **MDS:** myelodysplastic syndrome; **MM:** multiple myeloma; **NSCLC:** non-small cell lung cancer; **RCC:** renal cell carcinoma.

## 8. Allogeneic γδ T Cell Adoptive Transfer Therapy

In almost all autologous studies, treatments showed some promise and reduced tumor burden in at least a few patients, but the effects were inconsistent. Researchers looked for a way to standardize the cell product: introducing allogeneic therapeutic cells harnessed from healthy donors into multiple patients (Table 3 and Table 4). In a pilot study, Wilhelm et al. took a partial step toward this by selecting donors who were half-matched (HLA-haploidentical) family members of four patients with advanced hematological malignancies [109]. These patients had been judged ineligible for allogeneic stem cell therapy due to its high toxicity. Prior to infusion, all patients underwent lymphocyte-depleting chemotherapy. They were then injected with 99% pure γδ T cells from the family donors, along with ZOL on day 0 and repeated IL-2 infusions afterwards to promote cell expansion. Neutropenia occurred within the expected time range, with hematopoiesis recovering several weeks after the chemotherapy, and no patient experienced GvHD or organ injury. Patients experienced a mean 68-fold increase of donor γδ T cells, peaking around day eight, but persisting up to 28 days within the body. One patient passed away due to an infection unrelated to treatment, but the remaining three patients all experienced complete remission as a result of the therapy. This pilot study indicated that the use of allogeneic γδ T cells is feasible and safe, and that ZOL/IL-2 infusions can activate and expand allogeneic γδ T cells in vivo to achieve promising therapeutic response. Even with non-family donors, the safety of allogeneic γδ T cell therapy remained. A case study by Alnaggar et al. gave eight infusions of γδ T cells expanded from a healthy donor to a patient with cholangiocarcinoma over the course of one year [110]. The patient experienced no adverse effects. And even without ZOL or IL-2 supplementation, he experienced depleted tumor activity, tumor size reduction, and improved quality of life during and after the treatment.

## 9. CAR-Engineered γδ T Cell Adoptive Transfer Therapy

With clear evidence that adverse effects would be minor and reversible, and that γδ T cells had the potential to achieve a complete response, researchers genetically manipulated the cells to increase their antitumor functions. In one of the earliest pre-clinical studies of CAR-γδ T cells, Deniger et al. electroporated polyclonal γδ T cells with Sleeping Beauty (SB) transposon and transposase to overexpress a CD19 CAR and expanded them with artificial antigen-presenting cells (aAPCs) engineered to express CD19 [112]. CAR-γδ T cells and IL-2 were later intravenously injected into NSG mice xenografted with a CD19+ leukemia cell line. The group administered the cells and IL-2 weekly for three weeks, and results indicated that polyclonal CAR-γδ T cells can suppress tumor growth in this pre-clinical setting.

In 2020, Ang et al. expanded Vγ9Vδ2 T cells with Zometa/IL-2 and engineered the cells with a CAR based on the extracellular domain of NKG2D (NKG2DL) using mRNA electroporation [113]. The NKG2DL CAR-γδ T cells were capable of killing multiple solid tumor cell lines in vitro, including those resistant to Zometa treatment. In addition, they conducted pre-clinical studies in mice previously intraperitoneally injected with colon tumors or ovarian tumors and demonstrated NKG2DL CAR-γδ T cells caused tumor regression in mice and extended their survival in both tumor models.

Another group, Rozenbaum et al., expanded CD19 CAR-Vγ9Vδ2 T cells using ZOL/IL-2. They gave CD19+ tumor bearing NSG mice the therapeutic cells plus ZOL and observed tumor reduction [114]. Importantly, the group generated a CD19 knockout tumor model by CRISPR/Cas9 and showed that CD19 CAR-Vγ9Vδ2 T cells in the presence of ZOL had enhanced cytotoxicity against CD19- tumor cells compared to conventional CD19-CAR T cells. This important finding suggests that CAR-Vγ9Vδ2 T cells combined with a bisphosphonate compound could target tumors in an antigen escape situation through CAR-independent pathways, significantly increasing their therapeutic potential in clinics.

Table 4 summarizes past and current CAR-γδ T cell clinical trials. A phase I clinical trial (NCT02656147) to assess the safety and efficacy of CD19 CAR-γδ T cells for the treatment of lymphoma and leukemia was conducted by Beijing Doing Biomedical. Another phase I clinical trial (NCT04107142) performed by CytoMed Therapeutics has evaluated the safety and tolerability of haploidentical/allogeneic NKG2DL CAR-γδ T cells in patients with R/R solid tumors, including colorectal cancer, triple-negative breast cancer, sarcoma, nasopharyngeal cancer, prostate cancer, and gastric cancer. Although both studies have been completed, no formal reports on their results have yet been published. Currently, PersonGen BioTherapeutics is validating its CD7 CAR-γδ T cells in patients with R/R CD7+ T cell-derived malignant tumors in an early Phase I study (NCT04702841). Adicet Bio is testing CD20 CAR-Vδ1+ T cells (ADI-001) in patients with B cell malignancies in another Phase I trial (NCT04735471). In December 2021, Adicet Bio announced very positive interim clinical data from this study. Of the four efficacy evaluable patients, three received CAR-Vδ1+ T cells at a lower dose (30 million CAR+ cells) and one received a higher dose (100 million CAR+ cells). Among the low dose patients, one achieved a complete response (CR), one achieved a near CR, and one had progressive disease (PD). For the higher dose patient, a CR was achieved. This encouraging result granted the company’s lead candidate ADI-001 “Fast Track Designation” from the FDA.

## 10. Other γδ T Cell-Based Therapies

Efforts to optimize γδ T cell-based immunotherapies have resulted in a diversity of approaches to supplement the unique properties of γδ T cells, increase efficacy, and reduce side effects. One approach is harnessing a high affinity Vγ9Vδ2 TCR from γδ T cells and overexpressing it in αβ T cells. The engineered cell products (TEGs) possess the properties of αβ T cells, such as robust proliferation and longevity, with the additional targeting ability of the Vγ9Vδ2 TCR to BTN molecules on tumor cells without MHC restriction. After verifying these cells’ efficacy and safety in a PDX leukemia model [115], Gadeta began a Phase I clinical trial (NTR6541), assessing its lead candidate, TEG001, in patients who had exhausted other treatment options for B cell malignancies, high-risk Myelodysplastic Syndrome, or MM (Table 2). Conventional CAR constructs combine the CD3ζ intracellular signaling domain and co-stimulatory domain, enabling one antigen binding event to generate the two activating signals required for T cell stimulation. Fisher et al. sought to avoid overactivation of a therapeutic product against any healthy cells that may express a CAR target antigen [116]. They attempted to minimize reactions against healthy cells by GD2 CAR-γδ T cells by separating these activating signals. By replacing both CD3ζ and the CD28 co-stimulatory domain with DAP10 (the NKG2D co-stimulatory domain) in their CAR construct, two independent tumor binding events—Vγ9Vδ2 TCR signaling plus CAR signaling—were required for cell activation. This separated signaling method could develop safer immunotherapies with less potential for off-target effects, allowing broader target selection.

Additional immunotherapy approaches have attempted to improve targeting and efficacy by developing bispecific antibodies featuring two independent domains that link T cells to tumor cells. It is difficult to target an antibody to an inherently variable αβ TCR, so researchers primarily chose to target CD3 [117]. Two FDA-approved examples are blinatumomab, which targets CD3 and CD19 [118], and catumaxomab, which targets CD3 and EpCAM [119], and can also be recognized by CD16 on the γδ T cell membrane. However, because CD3 is ubiquitous among different TCRs, anti-CD3 antibodies can indiscriminately activate all T cell subsets and induce cytokine storms, causing significant adverse effects in the first few cycles of the drug. To reduce these side effects, finer effector cell selectivity is ideal. The consistent and unique molecules that comprise the γδ TCR present a steadier target for selective stimulation of one cell type. A bispecific antibody binding the Vγ9 chain and HER2 tumor antigen enhanced Vγ9Vδ2 T cell in vitro and in vivo killing of pancreatic adenocarcinomas, with superior in vitro killing compared to pAg stimulated γδ T cells alone or with a CD3/HER2 bispecific antibody [120]. An additional report with a Vγ9/CD123 bispecific antibody in acute myeloid leukemia (AML) provided further support for a Vγ9 targeting strategy over a CD3 targeting strategy [121]. This bispecific antibody selectively stimulated Vγ9+ T cells, with significantly less production of cytokine storm associated IL-6 and IL-10 compared to a CD3/CD123 antibody. The Vγ9/CD123 antibody mediated γδ T cell conjugation with tumor cells, and induced cytotoxicity against antigen expressing tumors both in vitro and in vivo. Currently, Lava Therapeutics is conducting a Phase I/II trial (NCT04887259) for LAVA-051, an antibody which binds both the Vδ2 chain and CD1d, allowing for selective stimulation of Vγ9Vδ2 T cells and NKT cells to target blood cancers such as chronic lymphocytic leukemia (CLL), MM, and AML (Table 1). In their preliminary data, the Vδ2/CD1d antibody was well-tolerated without cytokine storm syndrome and circulating Vγ9Vδ2 T cells effectively engaged the antibody and displayed a more activated phenotype after treatment.

Other than the approaches discussed above, stem cell derived γδ T cells for adoptive transfer cell therapy could be another promising direction. Watanabe et al. generated induced pluripotent stem cells (iPSCs) that harbor rearrangements of the TCR gamma chain (Vγ9) and TCR delta chain (Vδ2) gene regions (γδT-iPSCs) [122]. They activated PBMC γδT cells with IL-2 and ZOL, then transduced those cells with the Sendai virus vector encoded KLF4, c-MYC, OCT3/4, and SOX2 transcription factor genes. They successfully reprogrammed the cells into iPSC lines that were pluripotent and capable of differentiating into embryoid bodies (EB) and hematopoietic progenitor cells. However, they did not continue to differentiate the cells into fully mature γδT cells. Zeng et al. also reprogrammed γδT-iPSC lines from Vγ9Vδ2 T cells [123]. They further differentiated the cells into NK-like γδ T cells, which they designated “γδ natural killer T” (γδ NKT) cells. The group used a “NK cell-promoting protocol” and utilized OP9 mouse stromal cells overexpressing DLL1 Notch ligands to support the differentiation process. The iPSC derived γδ NKT cells exhibited a NK cell-like phenotype and expressed many NK receptors such as NKG2D, DNAM-1, NKp30, NKp44, NKp46, and CD16. An in vitro cytotoxicity assay demonstrated that these cells are powerful killers of various cancer cell lines, especially solid tumors.

## 11. γδ T Cell-Based Cancer Immunotherapy: Challenges and Perspectives

The past several decades of immunology research have yielded great progress in our understanding of the molecular mechanisms underlying antigen presentation to αβ T cells and their subsequent response. However, progress in understanding these mechanisms in γδ T cells has been comparatively slow and is impeded by limited knowledge of what antigens these cells respond to [124]. Vγ9Vδ2 T cells were initially thought to directly recognize pAgs produced by microbial pathogens and tumor cells with abnormal MVA pathway activation [125]. Increasing structural and biochemical evidence evolved this understanding, indicating instead that the Vγ9Vδ2 TCR reacts to BTN3A1 on the cell surface, which is bound at its intracellular B30.2 domain by pAgs [24]. More recent evidence has refined this paradigm, proposing that a BTN2A1-BTN3A1 complex acts as a dual ligand binding the Vγ9Vδ2 TCR in response to intracellular pAg binding the B30.2 domain [27]. A better understanding of mechanisms of γδ T cell activation and its TCR targeting may allow us to maximize the potential of γδ T cell-based therapy. While certain subsets of γδ T cells have been selected as potential immunotherapy agents, other subsets were identified as pro-tumoral γδ T cells and have immunosuppressive roles in some cancers [126,127]. Peng et al. showed that a dominant Vδ1 T cell population infiltrating breast tumors were associated with immunosuppressive functions by inhibiting naive and effector T cell responses and by impairing DC maturation and function [128]. In 2014, Wu et al. reported IL-17-producing γδ T cells were found to have a pro-tumoral role [129]. In particular, Vδ1 T cells were the primary source of IL-17 in colorectal cancer chronic inflammation. Furthermore, these Vδ1 T cells produced TNF, IL-8, and granulocyte–macrophage colony-stimulating factor (GM-CSF), which attracted immunosuppressive myeloid-derived suppressor cells (MDSCs) into the tumor microenvironment to support immunosuppressive activity [129]. Besides these two reports, several studies also found that γδ T cells may have immunosuppressive roles in different cancers [130,131,132,133]. Although γδ T cells have tremendous potential for therapeutic applications in cell-based immunotherapy, their pro-tumoral mechanisms need to be studied further to prevent their possible deleterious functions for cancer treatment.

Another challenge of γδ T cell-based cancer immunotherapy is that although several studies have reported that γδ T cells can modulate immunosuppressive cells in the TME, the lack of nutrients, presence of inhibitory molecules, and hypoxia in the TME may still limit the therapeutic potential of γδ T cells [67,134]. Inhibitory molecules such as TGF-β [135], prostaglandin-E2 (PGE2) [136], adenosine [137], and soluble NKG2D ligands [138] produced by tumor cells and other cells of the TME can interfere with the proliferation and function of γδ T cells. Gonnermann et al. reported that Colo357 cells (pancreatic adenocarcinoma) develop resistance to γδ T cell lysis by enhancing the expression of cyclooxygenase 2 (Cox-2) [139]. This enzyme catalyzes the metabolism of arachidonic acid to prostanoids, such as PGE2, which in turn promotes tumorigenesis, tumor progression, and metastasis. This Cox-2 metabolite contributes to establishing immunosuppressive factors in the TME and inhibits the antitumor activities of immune cells. However, the PGE2-mediated inhibition of γδ T cell cytotoxicity can be partially overcome with Cox-2 inhibitors [139]. This suggests that the use of Cox-2 inhibitors with γδ T cells or other cell types may improve the cytotoxic functions of immune cells in the TME and have beneficial effects in clinics.

Exhaustion is a hyporesponsive cell state induced by chronic stimulation from repeated antigen exposure, and manifests as a loss of effector functions including proliferation, cytokine production, and cytotoxicity. These changes are concomitant with the expression of inhibitory receptors such as PD-1, TIM-3, LAG-3, and CTLA-4 [140,141]. Although exhaustion may have homeostatic function in limiting immunopathology in chronic infections, in immunotherapy it can be an obstacle reducing the efficacy of cell therapies. CD8+ T cell exhaustion has been documented in many cancers, and this process also occurs in γδ T cells [142]. In a nonhuman primate model, Vγ9Vδ2 T cells repeatedly stimulated with pAgs and IL-2 had lower expansion rates upon restimulation, compared to the initial pAg response. γδ T cells from CLL patients failed to proliferate in response to IPP accumulation in CLL cells, indicating an exhausted state that could arise from prolonged stimulation from these patients’ cancer cells [143]. The TME often has upregulated expression of ligands for inhibitory receptors present on immune cells, which engage to enforce an exhausted state. In one report, MM patients’ bone marrow (BM) had upregulated PD-L1 and contained Vγ9Vδ2 T cells with greater PD-1 expression than γδ T cells from other regions, as well as reduced proliferation response to ZOL [144]. This study found that an anti-PD-1 treatment conferred a fivefold increase in proliferation of MM BM γδ T cells after ZOL stimulation. Another study found that anti-PD-1 treatment increased γδ T cell IFN-γ production both with direct ZOL stimulation, and from ZOL-treated primary AML cells [145]. Using γδ T cells combined with immune checkpoint inhibitors is thus appealing to preemptively guard against exhaustion from sustained interactions with tumors, maintaining cell effector functions for a more efficacious therapy.

γδ T cells are especially prone to exhaustion when differentiated toward an effector memory (T_EM_) or terminally differentiated effector memory (T_EMRA_) state [146,147]. T cells progress to more T_EM_ and T_EMRA_ states in response to activation signals and stimulation, including tumor recognition [148,149]. γδ T cell memory states have traditionally been defined by the expression of CD45RA and CD27, as well as several lymph node homing receptors [147]. Naive γδ T cells are the earliest memory state. They are CD45RA+CD27+ and express the lymph node homing markers CD62L and CCR7, and are infrequent in circulation but are the predominant γδ T cell memory type in lymph nodes [147]. Naive γδ T cells proliferate in response to pAg but have minimal cytokine releasing activity. These cells have strong self-renewal potential to yield a population with the same memory state, a property which diminishes with greater differentiation. After stimulation, naive γδ T cells remain CD62L+ and CCR7+ but progress to a CD45RA-CD27+ central memory (T_CM_) state that is CD45RO+ and mostly found in the peripheral blood [147]. T_CM_ γδ T cells have greater proliferative potential in response to pAg compared to naive cells, but still have reduced effector functions such as cytokine release. Sites of inflammation contain further differentiated γδ T cells without CD62L or CCR7, including CD45RA-CD27- T_EM_ cells and CD45RA+CD27- T_EMRA_ cells. T_EM_ γδ T cells have reduced proliferative potential and minimal self-renewal ability, but vigorously release cytokines including IFN-γ. Although T_EMRA_ γδ T cells exert potent cytotoxicity against tumor cells, they soon become exhausted when they are excessively restimulated by tumor antigens. Exhausted T cells not only lose the ability to proliferate and self-renew, but also have weakened antitumor functions [150].

γδ T cell expansion protocols induce γδ TCR signaling to yield practical cell numbers at the expense of the population’s memory state. While the original γδ T cells being expanded may contain some γδ T cells in earlier memory states, stimulation differentiates the resultant population toward effector and terminally differentiated effector memories. Therapeutically, these further differentiated populations could initially yield potent antitumor effects, but longer-term tumor control would be limited by their low proliferation/self-renewal ability and persistence. Instead, maintaining the naive or central memory status of γδ T cells is desirable for clinical use. These populations have long persistence, which is amenable for ongoing tumor surveillance; upon antigen stimulation they proliferate greatly to yield a large population of tumor responsive effector cells while reserving a subset of self-renewing memory cells to generate future waves of effectors against subsequent tumor encounters. Since current γδ T cell expansion protocols rely on stimulatory signals which induce differentiation away from these memory states, generating a population of naive and central memory γδ T cells in vitro large enough for therapeutic use could be challenging. However, such direction is crucial for therapeutic efficacy. Fortunately, there have been continuous advances in knowledge about γδ T cell biology and culture conditions. As γδ T cells increasingly attract research attention for their immunotherapeutic potential, growing bodies of work in these areas could yield key insights to effectively generate γδ T cells that have superior persistence and efficacy against tumors.

## Figures and Tables

**Figure 1 cancers-14-03005-f001:**
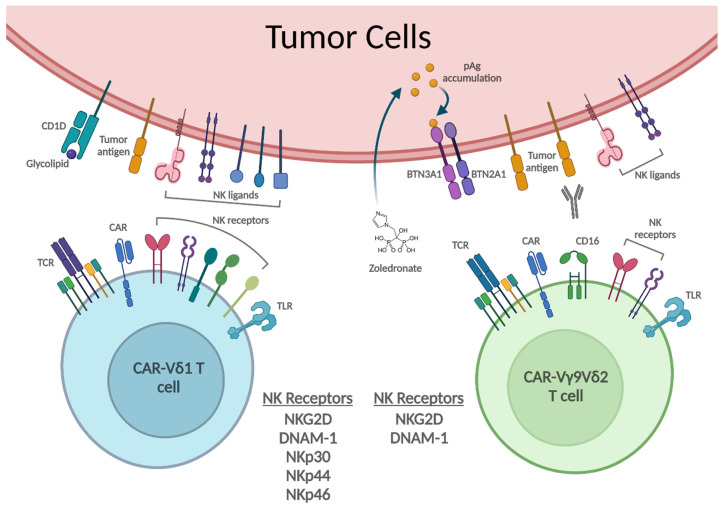
The tumor cell recognition of CAR-γδ T cells. CAR-γδ T cells recognize tumor cells through multiple receptor pathways, increasing their cytolytic potential and resistance to immunosuppression. Both Vδ1+ and Vγ9Vδ2 TCR possess MHC-nonrestricted cytotoxicity. The Vδ1 TCR recognizes lipid antigens presented on CD1 molecules, and the Vγ9Vδ2 TCR recognizes intracellular pAg-bound BTN3A1/2A1 complex. Vγ9Vδ2 T cells express CD16 (FcγRIII), which binds the Fc region of IgG antibodies to induce antibody-dependent cellular cytotoxicity (ADCC). TLRs (Toll-Like Receptors) form another axis of immune cell coordination with each binding pathogen- or damage-associated molecular patterns (PAMPs/DAMPs), simultaneously increasing γδ T cell cytotoxicity and coordinating with surrounding dendritic cells (DCs). NK receptor pathways expressed on the cells enable cytolytic activity in response to expressed markers of DNA damage, stress, or infection. Both Vδ1+ and Vγ9Vδ2 T cells express NKG2D, which recognizes cellular distress ligands presented on MIC(A/B) and ULBP1-6, and DNAM-1, which binds CD155 and CD112. In addition, Vδ1+ T cells express NKp30, NKp44, and NKp46.

**Figure 2 cancers-14-03005-f002:**
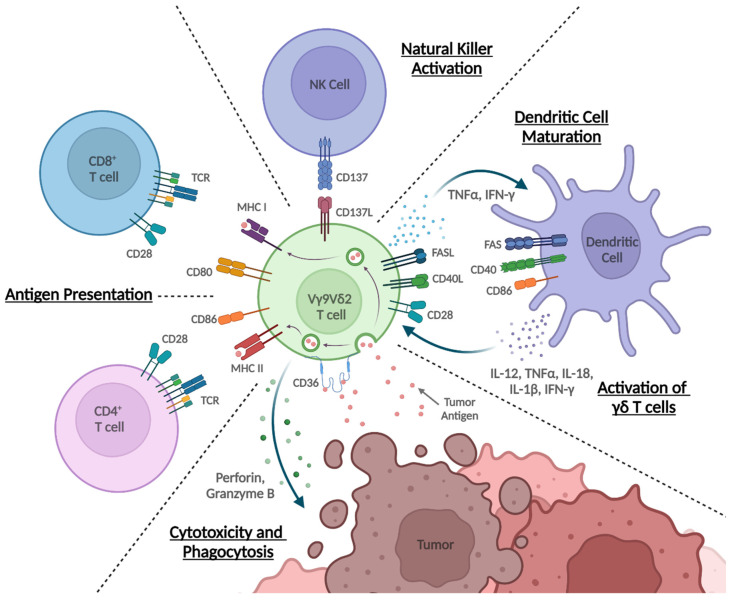
Regulation of antitumor immunity by Vγ9Vδ2 T cells. Vγ9Vδ2 T cells perform diverse immunological functions through interactions with other cells. They are directly cytotoxic to tumor cells through multiple mechanisms including perforin and granzyme B production, and express CD36 to facilitate tumor antigen phagocytosis. Upon incorporating antigens, Vγ9Vδ2 T cells act as antigen presenting cells (APCs) and both directly and cross present antigens on MHC molecules to αβ T cells. Combined with expression of the co-stimulatory molecules CD80 and CD86, Vγ9Vδ2 T cells effectively induce αβ T cells to differentiate and proliferate. CD137L expressed on Vγ9Vδ2 T cells activates NK cells by engaging CD137, increasing their NKG2D expression, direct cytotoxicity, and ADCC activity. Vγ9Vδ2 T cells induce DC maturation through TNFα and IFN-γ production, and engage CD40 to allow DCs to generate CD8+ T cell responses. Conversely, DCs activate Vγ9Vδ2 T cells by providing CD86 co-stimulation and producing cytokines including IL-12, IL-18, IL-1β, TNFα, and IFN-γ. Furthermore, mature DCs treated with ZOL upregulate CD40L on Vγ9Vδ2 T cells.

**Table 3 cancers-14-03005-t003:** Allogeneic, ex vivo expanded γδ T cell clinical trials.

Year	Author	Phase	Tumor Type	Effector Cells	Clinical Outcome
2014	Wilhelm et al. [109]	Pilot clinical study	NHL, MM, AML, PCL	Vγ9Vδ2 T, with some CD4-/8- αβT + ZOL + IL-2	** *CR: 3/4* ** - 1 patient dropped out of the study. - Patients experienced an average 68-fold expansion of donated Vγ9Vδ2 T cells.
2019	Alnaggar et al. [110]	Case Study	Cholangiocarcinoma	Vγ9Vδ2 T, 8 infusions	** *CR: 1/1* ** - Lack of adverse effects.
2020	Lin et al. [111] (NCT03180437)	I/II	Pancreatic cancer	IRE + Vγ9Vδ2 T	- Patients receiving multiple courses of γδ T cell infusion had a statistically significant extension in length of life.

**Abbreviations** are as follows: **CR:** complete response/remission; **ZOL:** zoledronate/zoledronic acid; **IRE:** Irreversible electroporation; **AML:** acute myeloid leukemia; **MM:** multiple myeloma; **NHL:** non-Hodgkin’s lymphoma; **PCL:** plasma cell leukemia.

**Table 4 cancers-14-03005-t004:** Allogeneic, ex vivo expanded CAR-γδ T cell clinical trials.

Year	Author	Phase	Tumor Type	Effector Cells	Clinical Outcome						
2017	Beijing Doing Biomedical (NCT02656147)	I	B-Cell Lymphoma, ALL, CLL	CD19 CAR-γδ T	-						
2019	CytoMed Therapeutics (NCT04107142)	I	Solid tumors	NKG2DL CAR-γδ T	-						
2021	Adicet Bio (NCT04735471)	I	B-cell malignancies	CD20 CAR-Vδ1 T + IL-2	Interim result–***CR: 2/4, PR: 1/4, PD: 1/4***						
2021	PersonGen BioTherapeutics (NCT04702841)	I	CD7+ T lymphoma (ALL)	CD7 CAR-γδ T	-						

**Abbreviations** are as follows: **PD:** progressive disease; **PR:** partial response; **CR:** complete response/remission; **CAR:** chimeric antigen receptor; **ALL:** acute lymphocytic leukemia; **CLL:** chronic lymphocytic leukemia.

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
