# Peer review of "Human γδ T Cell Subsets and Their Clinical Applications for Cancer Immunotherapy"

_cancers, 2022, doi:10.3390/cancers14123005_

Round 1

Reviewer 1 Report

The review manuscript "Human γδ T Cell Subsets and Their Clinical Applications for Cancer Immunotherapy" introduced the features of different subsets of human γδ T cells and their functions in antitumor immunity. The authors summarized past and ongoing clinical studies utilizing γδ T cell-based cancer therapy, while also pointing out challenges and perspectives in this field. This review provides a comprehensive overview of the function of γδ T cells and offer future directions that may improve cancer treatments using this T cell subset. I recommend that this manuscript be accepted with the following minor revisions:

Minor comments

1.      The authors could create a table to relate the γδ T cell subset with their roles in TME and their therapeutic use.

2.      In the main text, Figure 2 is referred earlier than Figure 1. The authors could swap the order of the two figures.

3.      I don’t think the first sentence of the manuscript, referring to the current limitations of CAR-T therapy is necessary - and may not age well. It would be preferable to use this first sentence to refer to the prior success and promise of various T cell therapies (CAR-T, TCR-T, and TIL) as a launching point for describing γδ T cells and why they may be important. The points about current limitations can be explored later in the paper.

4.      Line 47: these cells are “MHC-unrestricted”, but it is not clear what antigen-presenting molecules (APMs) are being recognized. Do they all recognize CD1 proteins or other APMs? This should be clarified here.  

5.      Remove the extra periods from headings (ex., in Line 55).

Author Response

1. We did not provide a table to relate the γδ T cell subset with their roles in TME and their therapeutic use because we think there are too few studies. And currently only Vδ2 T cells have some relevant studies, while other subsets do not.

2. We have swapped the order of the two figures. Figure 2 is now Figure 1, and Figure 1 is now Figure 2.

3. We also edited the first couple sentences of the article by addressing the success and promise of CAR-T cell therapy, and provided the rationale why we should consider innate T cells and gamma delta T cells as the CAR carrier. 

4. We also fixed the “MHC-unrestricted” sentence by clarifying how gamma delta T cells recognize their targets.

5. We have removed extra periods from headings (those periods were not there from the version we had, perhaps they were caused by using different versions of Microsoft Word to open the article?)

Reviewer 2 Report

This article is well constructed and the viewpoint is excellent. I decide that this article is acceptable in our journal.

Author Response

Thank you so much! :)

Reviewer 3 Report

The review entitled “Human γδ T Cell Subsets and Their Clinical Applications for Cancer Immunotherapy “is well written and pleasant to read. The review provides information on γδ T-cell biology and their importance in anti-tumor immune response and immunotherapeutic strategies.  The following points should be considered:

1-      Authors did not mention the frequency of γδ T-cell in solid tumors compared conventional T-cells and/or NK cells for instance. It would be interesting to evaluate (depending on cancer types) if an increased tumor infiltration of γδ T-cell correlates with good or bad prognosis of cancer patients and/or response to conventional therapies.

2-      Mechanisms of γδ T-cell activation by recognition of phospho-antigens and lipid antigens are well described. However, authors didn’t describe how mevalonate pathway is dysregulated in tumor cells compared to normal cells. Are pAg found to be increased in tumor cells?  Are tumor cells characterized by mutations of genes encoding (i) enzymes involved in mevalonate pathways (i.e. farnesyl pyrophosphate synthase) and/or (ii)  SREBP transcription factors thus causing pAg accumulations?

3-      It should be interesting to mention if pro-tumor function of γδ T-cell is described in literature

4-      Could you please indicate clinical trial numbers in table 1-4?

Author Response

1. We think we did mention about “the frequency of γδ T cells in solid tumors compared conventional T cells and/or NK cells” in our article. In line 47, Gentles et al…., this reference (#2) studied the frequencies of intratumoral T cells, γδ T cells, and NK cells and concluded γδ T cells were the best predictors of patient survival.

2. We have updated the mechanism of mevalonate pathway in line 101-105. 

3. We wrote a new paragraph (line 540-554) in section of “γδ T Cell-Based Cancer Immunotherapy: Challenges and Perspectives” to discuss about pro-tumoral potential of γδ T cells.

4. We tried our best to locate clinical trial numbers, but some of them were not found. We have included the trial numbers we found and please let us know it’s better to have them or not. Most of γδ T cell review papers do not have clinical trial numbers in their tables...